Manuscript prepared for Geosci. Model Dev.
with version 2015/11/06 7.99 Copernicus papers of the LaTeX class copernicus.cls.
Date: 17 May 2016

# Improved Forecasting of Thermospheric Densities using Multi-Model Ensembles

Sean Elvidge[1], Humberto C. Godinez[2], and Matthew J. Angling[1]

[1]Space Environment and Radio Engineering Group, University of Birmingham, United Kingdom
[2]Los Alamos National Laboratory, NM, USA

*Correspondence to:* Sean Elvidge (s.elvidge@bham.ac.uk)

**Abstract.**

This paper presents the first known application of multi-model ensembles to the forecasting of the thermosphere. A multi-model ensemble (MME) is a method for combining different, independent, models. The main advantage of using an MME is to reduce the effect of model errors and bias, since

it is expected that the model errors will, at least partly, cancel. The MME, with its reduced uncertainties, can then be used as the initial conditions in a physics-based thermosphere model for forecasting. This should increase the forecast skill since a reduction in the errors of the initial conditions of a model generally increases model skill. In this paper the Thermosphere-Ionosphere-Electrodynamic General Circulation Model (TIE-GCM), the US Naval Research Laboratory Mass Spectrometer and

Incoherent Scatter radar Exosphere 2000 (NRLMSISE-00) and Global Ionosphere Thermosphere Model (GITM) have been used to construct the MME. As well as comparisons between the MMEs and the 'standard' runs of the model, the MME densities have been propagated forward in time using TIE-GCM. It is shown that thermospheric forecasts of up to 6 hours, using the MME, have a reduction in the root mean square error of greater than 60%. The paper also highlights differences in

model performance between times of solar minimum and maximum.

## 1   Introduction

### 1.1   Background

NASA predicts that, by 2030, orbital collisions could become frequent enough to cause a cascade (Kessler et al., 2010), with the potential to prevent the use of low Earth orbit (LEO) (Koller, 2012).

One way to prevent a Kessler cascade is to more accurately predict orbital trajectories to better plan satellite collision avoidance manoeuvres. A key component in orbital trajectory predictions is an accurate description of the upper atmosphere, in particular the thermosphere, since drag due to atmospheric density is one of the main forces that affect the orbit of satellites and space debris. The neutral air density from 200 to 1000 km altitude (LEO) can change by 80% diurnally as well

as by at least one to two orders of magnitude during geomagnetic storms; sometimes in just a few

hours (Sutton et al., 2005; Lei et al., 2010). The upper atmosphere forecast models currently in use for orbit prediction are empirical and include NRLMSISE-00, the Jacchia Reference Atmosphere (Jacchia, 1977) and the NASA/MSFC Global Reference Atmospheric Model-1999 Version (Justus and Johnson, 1999). They are finely tuned, but when applied to satellite orbit forecasts they can result in large uncertainties in the orbital parameters. Often resulting in positional errors on the order of kilometres after a day (McLaughlin et al., 2011; Vallado and Finkleman, 2008).

One way to decrease the errors in satellite orbit forecasts is to reduce errors in thermospheric density forecasting. It has been previously suggested that ensemble modelling could improve space weather forecasts (Schunk et al., 2014). In this paper, multi-model ensembles (MMEs) are shown to enhance forecasts of the thermospheric density. The main objective is to minimize the prediction errors and bias of the forecasts by improving the initial conditions of the model.

## 1.2  Multi-Model Ensembles (MMEs)

The idea of improving model forecasts by combining two or more independent models is based upon a short note by Thompson (1977). Since then, MMEs have been extensively used in the climatology community with great success. For example, Doblas-Reyes et al. (2000) showed that using three climate models in an MME improved forecast skill. They also noted that the mean of an ensemble of forecasts has a smaller mean square error than any individual forecast. Evans et al. (2000) showed that the use of MMEs for both deterministic and probabilistic climate forecast verification significantly outperformed the individual models. Rozante et al. (2014) showed that an MME approach to forecasting had smaller root mean square errors (RMSEs) than any of the constituent models for most variables across their whole test scenario.

An MME relies on the idea that model forecasting can be improved by combining independent models (Thompson, 1977) and thereby reducing the impact of errors from individual models. Model errors arise in a variety of forms and include computational errors in physics model solvers (Rozante et al., 2014). For example, many physical systems can be described by a series of partial differential equations. Yet, in order to solve them, they have to be reduced to finite-dimensional ordinary differential equations to be integrable on a computer. Whilst necessary, this reduction introduces inaccuracies. Ridley et al. (2010) showed that solving the magnetohydrodynamics (MHD) equations numerically rather than analytically can cause significant differences in global MHD code. The same piece of code can give very different results by simply altering the numerical settings.

It is clear that an MME cannot give a result better than the best individual model in all circumstances. For a hypothetical perfect model of a system forming an MME will always add worse information. However in reality such perfect models do not exist and a successful MME should use independent, skilful, models. It is important to use independent models since models with similar error characteristics can find such characteristics amplified in the MME. It is impossible for the MME to be worse than all of the individual models (Hagedorn et al., 2005). However, if one model

is shown to consistently perform less well than all other models, then this should be excluded from the MME as it does not add useful information.

Although an MME may reduce the reported thermospheric density errors; it cannot alone forecast densities and thus cannot be directly used to improve satellite orbit forecasts. Errors in the forecasts given by thermospheric models are due to approximations in the modelled physics and uncertainties in the initial and boundary conditions. Pawlowski and Ridley (2009) showed that using different parameters within a global ionosphere-thermosphere model can cause differences in the reported temperatures and densities. As such some biases are expected in the models due to uncertainties in the parameters. Since MMEs are expected to reduce errors in the densities, these improvements can be used as the initial conditions for a forecast run of a physics model. Reducing the errors in the initial conditions is then expected to reduce the errors in the forecasted thermospheric densities. This paper will explore the effectiveness of MMEs using both the a posteriori knowledge and as an initial condition in a forecast model run.

One can construct an MME using a variety of different approaches, but they fall into two main categories, equal and unequal weightings.

### 1.3 Equally Weighted MMEs

There are a number of difficulties in constructing an MME. These include how the models should be combined and the fact that different models do not all share common output variables. A further problem is that there may not be observational data for each parameter, making it difficult to assess model performance for all parameters. One way to resolve the latter problem is to not take model performance into account and use an equally weighted average. Such a simple method for MME generation has been shown to increase model skill in climate studies. For example, Christensen et al. (2010) found that using a variety of different weight schemes for the construction of the MME did not provide consistent superiority over a simple averaging approach. Using a small dataset, Weisheimer et al. (2009) commented that finding a robust weighting system was difficult and suggested applying equal weights to the models. Their approach, using five models in the MME, lead to a significant improvement in seasonal-to-annual climate forecasts compared to any one individual model.

### 1.4 Weighted MMEs

Alternatively, the MME can use different weights for each model. There are different approaches for estimating the weights to be applied to individual models. These include a least-squares minimization of differences between the model and observations (Krishnamurti, 1999), a best linear unbiased estimate (BLUE) (Pavan and Doblas-Reyes, 2000) and a weighting scheme based upon the maximisation of a posterior likelihood function (Rajagopalan et al., 2002). All of the approaches depend on some measure of model skill. An appropriate skill measure must be chosen for each particular use of the MME. For example, Tebaldi and Knutti (2007) state that the skill of a (climate) model

should not be judged from its ability to predict the future, but instead from its ability to predict mean conditions, variability, and transient changes.

In the absence of existing MME work in the thermospheric literature, a sample mean square error (MSE) has been used in this work:

$$\text{Skill} = \text{MSE} = (\mu^2 + \sigma^2). \tag{1}$$

Where $\mu$ is the mean of the time series of errors (model minus observation) and $\sigma$ is the standard deviation of the errors. Weigel et al. (2008) has previously shown that the MSE can effectively be used in a weighting scheme for MMEs to increase climate model forecast skill.

## 2   Models and Observations

For this study three atmospheric density models have been used: NRLMSISE-00, TIE-GCM and GITM. NRLMSISE-00 is an empirical density model whereas GITM and TIE-GCM are physics-based models. The models are driven using standard geophysical indices: i.e. F10.7, which is the solar flux at a wavelength of 10.7 cm at the Earth's orbit and is used as a proxy for solar output and Kp or Ap, which indicate the severity of the magnetic disturbances in near-Earth space. Physics models of the ionosphere-thermosphere often suffer from biases. These can usually be attributed to the uncertainties in the model parameters which have a large impact on the final results (Pawlowski and Ridley, 2009). These biases can be reduced by modifying particular parameters. For example Burrell et al. (2015) showed that changing the photoelectron heating in GITM moves the baseline up and down. For this study the models are compared to observations from the CHAMP satellite. Each model and the CHAMP data is described in the following sections.

### 2.1   NRLMSISE-00

The US Naval Research Laboratory Mass Spectrometer and Incoherent Scatter radar Exosphere 2000 (NRLMSISE-00), is a global, empirical model of the atmosphere. It uses the 81 day average of F10.7, the daily F10.7 solar flux value of the previous day, and 3-hourly Ap to model the density and temperature of atmospheric components (Picone et al., 2002). It is based on the earlier MSIS-86 (Mass Spectrometer and Incoherent Scatter radar 1986) (Hedin, 1987) and MSISE-90 (Mass Spectrometer and Incoherent Scatter radar Exosphere 1990) (Hedin, 1991) models.

The model outputs number densities of helium, atomic oxygen, molecular oxygen, atomic nitrogen, molecular nitrogen, hydrogen and argon, as well as total mass density and the temperature at a given altitude. NRLMSISE-00 has been shown to offer a noticeable improvement over MSISE-90 (Picone et al., 2002) and Jacchia-70 (Jacchia, 1977).

## 2.2 TIE-GCM

The National Center for Atmospheric Research (NCAR) Thermosphere Ionosphere Electrodynamics - General Circulation Model (TIE-GCM) is a three-dimensional model of the coupled thermosphere ionosphere system (Richmond et al., 1992). At each time step the continuity, energy and momentum equations are solved for neutral and ion species using a fourth-order, centred finite difference scheme (Roble et al., 1988). TIE-GCM has two different grid settings: single and double resolution. The latitude values range from -87.5 to 87.5 in 5° steps at single resolution and 2.5° at double. In longitude it ranges from -180° to 180°. Altitude is calculated in pressure levels with half scale height for single resolution and quarter scale height for double. These correspond to heights from approximately 95 km to 550 km. For this work the single resolution grids have been used.

The model takes as input the daily F10.7, the 81 day F10.7 average and the Ap. It uses either the Weimer or Heelis models for the ionospheric electric fields at high latitudes (Heelis et al., 1982; Weimer, 2005). Throughout this work, the Heelis model has been used. The lower boundary condition (atmospheric tides) are given by the Global Scale Wave Model (GSWM) (Hagan et al., 1999).

## 2.3 GITM

The Global Ionosphere Thermosphere Model (GITM) is a physics-based three-dimensional global model that solves the full Navier-Stokes equations for density, velocity, and temperature for a number of neutral and ion species (Ridley et al., 2006). The model also provides the total neutral density, electron density, electron, ion and neutral temperatures, neutral wind speed and plasma velocities. For inputs, GITM uses F10.7 solar flux, hemispheric power (Emery et al., 2008) (available from the National Oceanic and Atmospheric Administration (NOAA) website (U.S. Dept. of Commerce, NOAA, 2015)), interplanetary magnetic field (IMF) data and solar wind velocity. The model allows the user to select latitude and longitude grids and uses a static altitude grid for the height profile which is set at initialization. For this work 5° grids have been used to coincide with the TIE-GCM grids.

To solve the continuity, energy and momentum equations, GITM uses an advection solver, whilst the ion momentum equation is solved assuming a steady state (Ridley et al., 2006). GITM inherently allows for non-hydrostatic solutions to develop which allows for realistic dynamics in the auroral zones (Ridley et al., 2006).

## 2.4 CHAMP

The performance of each model is compared against the atmospheric density fields derived from the CHAllenging Minisatellite Payload (CHAMP) satellite (Reigber et al., 2002). CHAMP was in operation from July 2000 to September 2010 and the reported neutral densities are derived from accelerometer data (Sutton, 2009). CHAMP was launched into a near polar orbit (87°) with an orbital

period of approximately 90 minutes. The initial altitude of the orbit was 454 km which decayed during the lifespan of the mission to 296 km by February 2010 due to atmospheric drag. Neutral densities were recorded approximately every 45 seconds. Accelerometer data was recorded every second and averaged such that neutral densities were reported approximately every 45 seconds.

## 3   Test Scenarios

Three separate test scenarios have been used during this study as (Table 1). For each of the test scenarios TIE-GCM and GITM were run for two days prior to the start date so that the 'spin up' period did not affect the final results. Test scenario 1 was chosen since it included a geomagnetic storm which took place on August 30th. The Ap index reached a high of 67 between 15UT and 18UT August 30th, whilst staying below 10 at other times. The F10.7 showed little variability throughout the whole test period (Figure 1). To further verify the results a second solar minimum test scenario was explored (scenario 2, November 19th to November 23rd 2008). Finally a solar maximum test (scenario 3) was also used. This test scenario also includes a large geomagnetic storm in the middle of the test period, where the Ap reached a high of 236.

## 4   Results

### 4.1   Initial Model Comparisons

To compare NRLMSISE-00, TIE-GCM and GITM with CHAMP, the output of each model was spatially mapped to the CHAMP position using trilinear interpolation. The model files were output every 30 minutes and the CHAMP observation closest to the model time was used. Figure 2 shows the modified Taylor diagram (Elvidge et al., 2014) for total neutral density for NRLMSISE-00, GITM and TIE-GCM compared to the CHAMP observations for each of the test studies. Figure 3 is the time series plot for the first test scenario (2009; solar minimum) of neutral density of the models and CHAMP for the same time period.

The NRLMSISE-00 empirical model results, as expected, show a reasonable mean approximation to the observed state, with the least bias of the tested models. However the model shows a larger variability in its output than the CHAMP observations. GITM shows a negative bias with a very small standard deviation compared to the observations (Figure 2), i.e. the range of values that GITM produces is smaller than the observations. GITM and NRLMSISE-00 have a very similar correlation, but GITM has the smaller error standard deviation. TIE-GCM has the strongest correlation of the models, but does show a positive bias and a standard deviation greater than that of the observation. TIE-GCM is also the only model to show some reaction to the storm. Although there is no increase in the maximum reported values, there is an increase in the minimum values (Figure 3).

The results from second test scenario (2008; solar minimum) are similar to the first (Figure 4).

TIE-GCM again has a positive bias and its standard deviation is greater than the CHAMP observations. TIE-GCM has a correlation to the CHAMP observations of ~0.5, the worst of the tested models. GITM and NRLMSISE-00 perform quite similarly in this test, albeit with GITM showing a negative bias. They have correlation coefficients compared to CHAMP of 0.87 and 0.89 (not statistically significantly different). Both models have normalised standard deviations less than unity (they

underestimate the range of observations).

Finally, the third test scenario (2001; solar maximum) has results which are considerably different to the other two test scenarios. The reported neutral densities compared to the CHAMP observations can be seen in Figure 5. The variability between the models, seen previously in Figure 3 and Figure 4, is greatly reduced in this test scenario. During quiet times the models all perform very similarly.

There is some variability in the models during the peak of the storm, with GITM in particular not responding as much as TIE-GCM or NRLMSISE-00. None of the models show any real bias, with a standard deviations close to the observations and similar error standard deviations. NRLMSISE-00 has the strongest correlation with the observations (0.73), whilst TIE-GCM and GITM are not significantly different (~0.5). This behaviour is to be expected since during solar maximum the

solar drivers, which are at a much higher level (Table 1), become dominant in the models. At solar minimum other internal and external dynamics dominate the evolution of the thermosphere densities. These other drivers are what cause the variability between the models in the other two test scenarios. It should be noted that the thermospheric densities during the extreme solar minimum of 2008/2009 were considerably lower than one would expect from the F10.7 levels (Solomon et al., 2010). This

could contribute to the poorer performance of the models compared to the CHAMP observations for the first two test scenarios.

The results from these test scenarios show that the models suffer from errors and biases, and are unable to exactly match the observed density field from CHAMP. In order to provide better forecasting abilities, MMEs can be used to combine the model output to minimize the impact of

model errors and bias.

### 4.2   Multi-Model Ensembles (MMEs)

As described in Sections 1.3 and 1.4 there are two approaches to constructing MMEs, simple averaging and more complicated weighting schemes. The mean square error has been chosen for the weighting scheme (Eq. 1) in this paper. The MSE was calculated using the models' neutral density

time series compared to the CHAMP observations. The model weights for the MME were then based upon the model skill. The inverse of the model skill was used to weight the models, so that the model

with the lowest MSE was weighted most heavily. That is, given the model skill of NRLMSISE-00, GITM and TIEGCM, $S_M$, $S_G$, and $S_T$ respectively, the weighting of model $i$ was calculated using:

$$\text{Weighting of model } i = \frac{1}{S_i \left( \frac{1}{S_M} + \frac{1}{S_G} + \frac{1}{S_T} \right)}. \tag{2}$$

The model skills and weighting of each model, for each test scenario, are given in Table 2. The large differences in model weightings between the different scenarios indicate that weighting the MMEs based on short-term historic performance is heavily dependent on the current conditions. Such an approach may not be suitable to forecasting.

For the first test scenario a further weighting scheme was used whereby before calculating the
MSE the model time series were restricted to times of low geomagnetic activity. Fuller-Rowell and Rees (1981) define quiet geomagnetic conditions as when the Kp index is between 0 and 1. In this study Ap values between 0 and 3 were used, which corresponds to a Kp of 0 to 1-. However, restricting the time series greatly reduces the number of data points (from 240 to 50). This means the weights may not be generally applicable to the full time series (Hagedorn et al., 2005). It should
be noted that the weightings used here are calculated using the same data set as is used in the test scenarios. In an ideal situation weightings should be calculated using a different (historic) data set and then used.

Figure 6 shows the neutral density time series of the observations, average and weighted MMEs for the first test scenario. Figure 7 is the modified Taylor diagram for the same test. It is clear that the
MMEs perform better than any of the individual models. The MME weighted across the whole time period performs the best of the MMEs. The MMEs all have little or no bias, and have a correlation to the CHAMP observations similarly to TIE-GCM. The all-times weighted MME in particular has a standard deviation close to the observations. The quiet-time weighted MME is the worst of the three MMEs and performs worse than the equally weighted MME. Therefore the quiet time weighted
MME was dropped from the analysis for the other two test scenarios.

The time series and modified Taylor diagram for the second test scenario are shown in Figure 8 and Figure 9. In this case the weighted MME performs as well as the best of the individual models (NRLMSIS-00 in this case). Although the equally weighted MME performs worse in terms of correlation compared to GITM and NRLMSISE-00, it still provides a significant improvement over
TIE-GCM and GITM in other regards (such as bias and standard deviation).

Figure 10 and Figure 11 show the results for the third test scenario. The MMEs have the same correlation as the best of the models (NRLMSISE-00) but also show a positive bias the weighted MME in particular. This is because NRLMSISE-00 itself has a large bias, but is heavily weighted (85.1%) in the MME. The MMEs offer some improvements in this test scenario (in correlation in
particular) but the improvement is not as pronounced as in the other scenarios. This is due to the dominant forcing of the solar drivers at solar maximum.

It has been shown, in these test scenarios, that combining model results leads to increased skill at matching the CHAMP derived data. In the following section this reduced uncertainty in atmospheric densities is used to provide the initial conditions of a forecast run of a model. Such an approach has been previously shown to increase climate model forecast skill (Tebaldi and Knutti, 2007).

### 4.3 Using the MME for Forecasting with TIE-GCM

The objective is to use the MME, with its reduced uncertainties, as the initial conditions for TIE-GCM. With the better initial conditions, it is expected that the forecast skill of TIE-GCM will be increased. In order to use an MME as the initial conditions for a physics-based model (i.e. TIE-GCM) more than just the combined neutral density is required. The MME of each density required by TIE-GCM (Table 3) has to be calculated. Where possible, the density for each model species required by TIE-GCM (e.g. oxygen; O) was found by combining the densities from NRLSMSE-00, GITM and TIE-GCM. However, for certain species (e.g. nitric oxide; NO) not all the models provide a density (in this case NRLMSISE-00). In these cases, just the models which do provide a density value were used. In cases where TIE-GCM has a density which no other model provides, the original data is used on its own. A similar approach is used for the temperatures and velocities.

To combine densities, temperatures and velocities from multiple models, the data must be interpolated to common latitude, longitude and altitude grids. Therefore NRLMSISE-00 and GITM grids were trilinearly interpolated to the TIE-GCM grid. The grids were then combined to form an MME. Since TIE-GCM uses pressure levels instead of altitude grids the MME values needed to be mapped back onto pressure levels. TIE-GCM provides a mapping between the pressure levels and geometric height for a given timestep. This mapping was used in reverse to morph the altitude grids to TIE-GCM readable pressure levels.

For the new TIE-GCM run, the model was restarted using the MME state-vector as the initial condition. TIE-GCM was then run for six hours with the model output recorded every 30 minutes. After the six-hour period, TIE-GCM was again restarted using the MME grid for the next six hour period. For the forecast run, the model only used the values of Kp and F10.7 corresponding to the initial conditions; i.e. they were not updated at each time step, but they were updated every six hours. This was so a true forecast could be simulated. The equally weighted MME uses no prior information so can be treated as true forecast. However, it should be noted that when using the weighted MME a true forecast is not obtained since the weighted MME is generated using the information from the CHAMP observations. Figure 12 is a flow chart of the process used to run TIE-GCM with the MME as its initial conditions for a six hour forecast, and Figure 13 is the procedure used for this test scenario.

Using the MME densities to initialize a run of TIE-GCM will alter the outputs of the model. It was expected that over time the two versions would converge. However this does not seem to happen over the six hour window used here. This is likely due to the fact that the model biases have a longer

time scale than six hours. Figure 14 shows the difference between a standard run of TIE-GCM and one started with the MME grid. It shows that the differences between the two models, started with different initial conditions, are decreasing towards zero, as expected, but it takes approximately 70 hours to reach these levels. The $e$-folding time of this is ~30 hours.

Figure 15 shows the modified Taylor diagram for neutral density compared to the CHAMP observations for the original TIE-GCM run, the NRLMSISE-00 (MSIS), GITM results and the results of rerunning TIE-GCM using the average and both weighted MME (all times and quiet times separately) as the initial condition every six hours for the first test scenario. Figure 16 is the reported time series of neutral densities from the CHAMP observations, the original TIE-GCM run and the results of rerunning TIE-GCM using the MMEs.

Using the MME densities as the starting point for TIE-GCM provides a clear improvement compared to the original run of TIE-GCM. The reported densities show very low bias and have variability close to the observations. In particular, the post-storm period is modelled very accurately in all but the quiet-time weighted MME. The average MME and all-times weighted initial conditions for TIE-GCM improves upon the original TIE-GCM correlation. Each of the TIE-GCM MME runs significantly improved the bias and all but the quiet-times improve the standard deviation of the model. The new TIE-GCM run (using the average MME) offers an improvement in all tested parameters compared to the neutral density MME calculated after the models were run (Figure 7). This is since the physics of one model, given initial conditions with lower errors, can propagate densities better than the average of three models, each with poor initial conditions.

None of the contributing models, nor the MMEs, model the peak of the storm period (~65 hours after August 28th 2009) with any accuracy. The best the models can do is to try and model the post-storm period as well as possible. This is because the models do not react quickly enough to the sharp increase in Ap (in terms of reported neutral densities).

The RMS error for each TIE-GCM MME run as well as the original TIE-GCM run compared to the CHAMP observations are shown in Table 4. The 95% confidence intervals are also reported. These have been calculated in the standard way,

$$
\left[ \sqrt{\frac{n}{\chi^2_{1-\frac{\alpha}{2},n}}} \text{RMSE}, \quad \sqrt{\frac{n}{\chi^2_{\frac{\alpha}{2},n}}} \text{RMSE} \right], \tag{3}
$$

where $n$ is the sample size, and $\alpha$ is the required confidence interval.

Figure 17 and Figure 18 show the results of using the MMEs as the initial conditions in TIE-GCM for test scenario 2. Again the improvements can be seen. The original TIE-GCM run had a correlation of ~0.5 with a large positive bias. However, when using the MME densities to initialize TIE-GCM a correlation of ~0.9 and no significant bias is achieved. The MME run provides results better than each of the constituent models. The reduction in RMS error between the original TIE-GCM run and the MME runs are shown in Table 4.

Finally Figure 19 and Figure 20 show the results for test scenario 3. In this case the NRLMSISE-00 model still provides the overall best results. However the TIE-GCM runs using the MME have less bias than NRLMSISE-00. The MME runs of TIE-GCM show an improvement in the post-storm modelling of neutral densities (Figure 20). The results from this test scenario once again highlight the weaknesses of this method for solar maximum conditions. Even when using the MME to initialize the model TIE-GCM still performs very similarly to when the conditions had not been changed. This is due to the dominance of the solar drivers.

It has been shown that the use of the MME as the initial conditions in TIE-GCM improve the models forecast skill considerably during solar minimum. The RMS error is reduced by approximately 60% ($\pm$ 6% for the 95% confidence interval). However no improvement to RMS error is achieved for the third test scenario (solar maximum).

## 5    Discussion and Conclusions

The work presented in this study shows the possibility of using multi-model ensembles (MMEs) to enhance the forecast skill of thermospheric models. Three models were used: an empirical model (NRLMSISE-00) and two physics-based models (TIE-GCM and GITM). The models' output density has been compared against derived density fields from CHAMP, where the models vary in performance compared to the observations depending on the test scenario. To improve the density estimation, an MME averaging technique has been applied and tested. Two approaches for the MME were used, a simple average MME where all models have the same weight, and a weighted MME, where each model is weighted according to its skill. Three different test scenarios have been used, two during solar minimum and one during solar maximum. The results show a significant improvement in both solar minimum cases. The MME was then used to initialize one of the physics-based models (TIE-GCM) to try and improve its forecast skill. During solar minimum test scenarios using the MME to initialize TIE-GCM shows a reduction in RMS error in neutral density of ~60% (Table 4). For solar maximum each of the models perform similarly and the MME provides no improvement to the model results. However it is important to note that the MME also does not degrade the results. It has been shown that using an equally weighted MME often provides as good, if not better, results than using a weighted MME. This is consistent with Hagedorn et al. (2005) who argued that for small data sets the most appropriate way to generate an MME is to use the unweighted average.

The results of this study show that the physics models suffer from large biases, as was discussed in Section 2. However these have been shown to not be systematic. For a given model they could be positive or negative, depending on the testing scenario. Burrell et al. (2015) argued that varying particular parameters can move the biases up and down easily. This approach should be used in the construction of future MMEs. If a number of model biases were of the same sign then these would likely end up contributing a bias to the MME. By running each model a number of times, with

varied input parameters, reduced-bias physics model results could be found. One could then extend the MME into a super-MME which contains both different models and different model settings. This is similar to the approach used by Palmer et al. (2000) in the climatology community who used nine instances of four different models in the construction of their MME.

Figure 14 showed that the MME started TIE-GCM run did not merge completely with the standard model run within 5 days. It takes over 70 hours for the two models to have zero differences in places. Further work should investigate the long-term model effects of the starting conditions for TIE-GCM (and physics models in general). Ridley et al. (2010) showed the influence of grid choice for global MHD code and it seems that something similar is happening with global ionospheric-thermospheric models.

A number of improvements could be implemented in generating the MME. Firstly, a separate 'training' data set should be used to generate the model weights to make a fairer test. A weighting scheme which varies based on longitude, latitude, height and time could also be implemented, as in Rozante et al. (2014)[Eq. 1]. A further approach would be to change the weighting scheme altogether and adopt Reliability Ensemble Averaging (REA) which is often used to generate MMEs in climatology studies (Giorgi and Mearns, 2002). In order to achieve this a larger number of models would be required. Also, in order to further verify the results longer test scenarios should be used to reduce the uncertainties in the statistics.

*Acknowledgements.* This research was, in part, conducted as part of the Integrated Modelling of Perturbations in Atmospheres for Conjunction Tracking (IMPACT) project at Los Alamos National Laboratory. More information is available at www.impact.lanl.gov. The work was commenced under the 3rd Los Alamos National Laboratory Space Weather Summer School. LA-UR-13-26825. The CHAMP data was collected from http://sisko.colorado.edu/sutton/data.html. TIE-GCM is developed by NCAR and is available at http://www.hao.ucar.edu/modeling/tgcm/tie.php. NRLMSISE-00 was developed by NRL and is available via the Community Coordinated Modeling Center (CCMC) at ftp://hanna.ccmc.gsfc.nasa.gov/pub/modelweb/atmospheric/msis/nrlmsise00/. GITM was developed at the University of Michigan and provided by Aaron Ridley. The authors also want to thank Aaron Ridley for his extremely helpful comments in the final construction of this paper.

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

**Table 1.** Test scenario descriptions. The CHAMP average altitudes and average F10.7 values are taken from across the 5 day test scenarios.

| Scenario Number | Start Time | Stop Time | CHAMP Average Altitude | Average F10.7 |
|:---:|:---:|:---:|:---:|:---:|
| 1 | August 28th 2009 | September 1st 2009 | 325 km | 68 |
| 2 | November 19th 2008 | November 23rd 2008 | 333 km | 69 |
| 3 | November 22nd 2001 | November 26th 2001 | 431 km | 179 |

**Table 2.** Model skill and associated weighting (calculated by the inverse of model skill, Eq. (2)) for use in the weighted MMEs for the three test scenarios.

| | Test Scenario 1 | | Test Scenario 2 | | Test Scenario 3 | |
|---|---|---|---|---|---|---|
| | Model Skill | Weight | Model Skill | Weight | Model Skill | Weight |
| **NRLMSISE-00** | $2.6 \times 10^{-24}$ | 49.0% | $3.08 \times 10^{-24}$ | 21.2% | $2.26 \times 10^{-25}$ | 85.1% |
| **GITM** | $4.52 \times 10^{-24}$ | 28.3% | $1.77 \times 10^{-24}$ | 36.9% | $3.01 \times 10^{-24}$ | 6.4% |
| **TIE-GCM** | $5.61 \times 10^{-24}$ | 22.7% | $1.56 \times 10^{-24}$ | 41.9% | $2.25 \times 10^{-24}$ | 8.5% |

**Table 3.** NRLMSISE-00, TIE-GCM and GITM model outputs. mmr is the mass mixing ratio.

| NRLMSISE-00 | TIE-GCM | GITM |
|---|---|---|
| He $(cm^{-3})$ | | He $(m^{-3})$ |
| O $(cm^{-3})$ | O (mmr) | O $(m^{-3})$ |
| $O_2$ $(cm^{-3})$ | $O_2$ (mmr) | $O_2$ $(m^{-3})$ |
| N $(cm^{-3})$ | N (mmr) | N $(m^{-3})$ |
| $N_2$ $(cm^{-3})$ | $N_2$ (mmr) | $N_2$ $(m^{-3})$ |
| $A_r$ $(cm^{-3})$ | | |
| H $(cm^{-3})$ | | H $(m^{-3})$ |
| | NO (mmr) | NO $(m^{-3})$ |
| | $O^+$ $(cm^{-3})$ | $O^+$ $(m^{-3})$ |
| | $O_2^+$ $(cm^{-3})$ | $O_2^+$ $(m^{-3})$ |
| | | $N^+$ $(m^{-3})$ |
| | | $N_2^+$ $(m^{-3})$ |
| | $NO^+$ $(cm^{-3})$ | $NO^+$ $(m^{-3})$ |
| | $N_e$ $(cm^{-3})$ | $N_e$ $(m^{-3})$ |
| Neutral temp. (K) | Neutral temp. (K) | Neutral temp. (K) |
| | Ion temp. (K) | Ion temp. (K) |
| | Electron temp. (K) | Electron temp. (K) |
| | Neutral meridional wind $(cms^{-1})$ | Neutral velocity (east) $(ms^{-1})$ |
| | Neutral zonal wind $(cms^{-1})$ | Neutral velocity (north) $(ms^{-1})$ |
| | Neutral vertical wind $(cms^{-1})$ | Neutral velocity (up) $(ms^{-1})$ |
| | | Ion velocity (east) $(ms^{-1})$ |
| | | Ion velocity (north) $(ms^{-1})$ |
| | | Ion velocity (up) $(ms^{-1})$ |
| | | O velocity (up) $(ms^{-1})$ |
| | | $O_2$ velocity (up) $(ms^{-1})$ |
| | | N velocity (up) $(ms^{-1})$ |
| | | $N_2$ velocity (up) $(ms^{-1})$ |
| | | NO velocity (up) $(ms^{-1})$ |

**Table 4.** The RMS error of the original TIE-GCM run and running the model with the MME as the inital conditions. The 95% confidence intervals are also reported.

|  |  | RMS Error $\times 10^{-12}$ (kgm$^{-3}$) | 95% Confidence Interval |
|---|---|---|---|
| **Test Scenario 1 (2009)** | Equal MME | 0.84 | [0.78, 0.92] |
|  | Weight-quiet MME | 1.2 | [1.1, 1.3] |
|  | Weight-all MME | 0.91 | [0.83, 1.0] |
|  | TIE-GCM Original | 2.4 | [2.2, 2.6] |
| **Test Scenario 2 (2008)** | Equal MME | 0.53 | [0.49, 0.59] |
|  | Weight-all MME | 0.53 | [0.49, 0.59] |
|  | TIE-GCM Original | 1.5 | [1.4, 1.7] |
| **Test Scenario 3 (2001)** | Equal MME | 1.2 | [1.1, 1.4] |
|  | Weight-all MME | 1.2 | [1.1, 1.4] |
|  | TIE-GCM Original | 1.3 | [1.1, 1.4] |

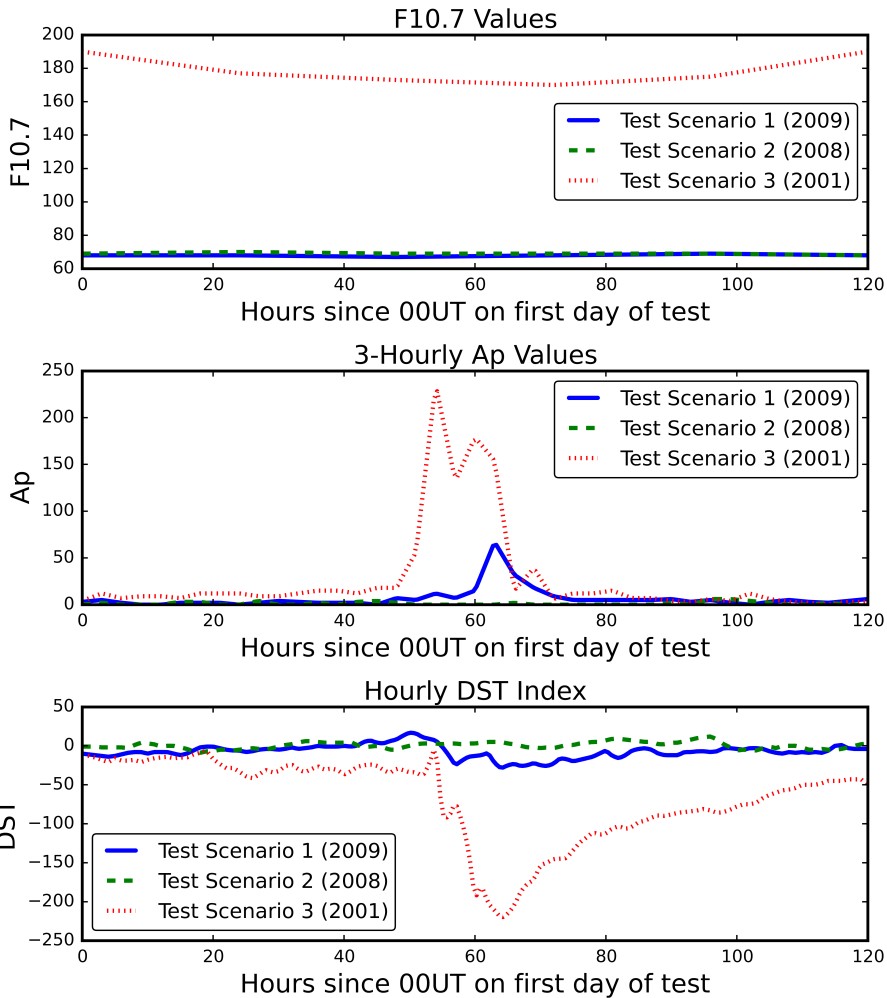

**Figure 1.** Ap, F10.7 and DST index values for the three test scenarios. The spikes in Ap for the 2009 and 2001 test scenario seem to be due to a geomagnetic storms.

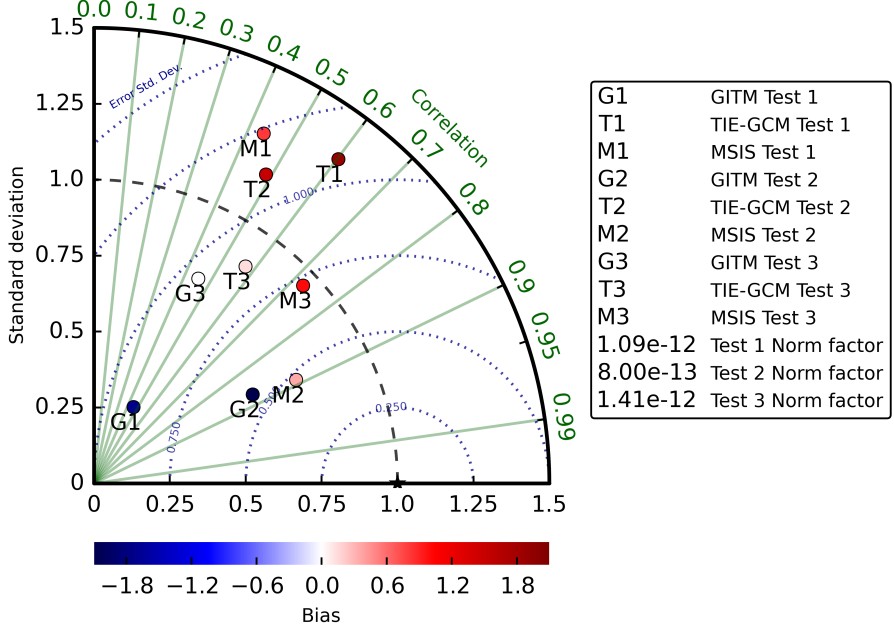

**Figure 2.** Modified Taylor diagram (Elvidge et al., 2014) for NRLMSISE-00 (MSIS), TIE-GCM and GITM for neutral density, compared with CHAMP for each of the three test scenarios. The azimuthal angle represents the correlation of the models neutral density time series with the CHAMP observation. The radial distance shows the standard deviation of the model time series and the semicircles, centred at a standard deviation of 1, is the standard deviation of the errors (model minus observation). The colour scale shows the bias (mean of model minus mean of truth). Each quantity is normalized and the original values can be reformed using the corresponding 'factor' in the top right of the diagram.

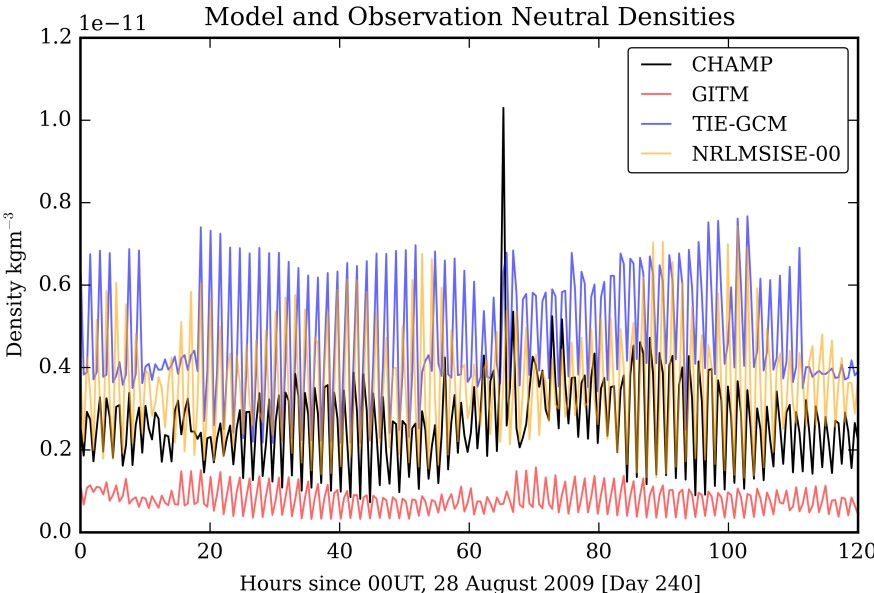

**Figure 3.** CHAMP, GITM, TIE-GCM and NRLMSISE-00 reported neutral densities for the first test scenario (2009; solar minimum). The fast oscillations are due to CHAMPs orbit (~90 minutes).

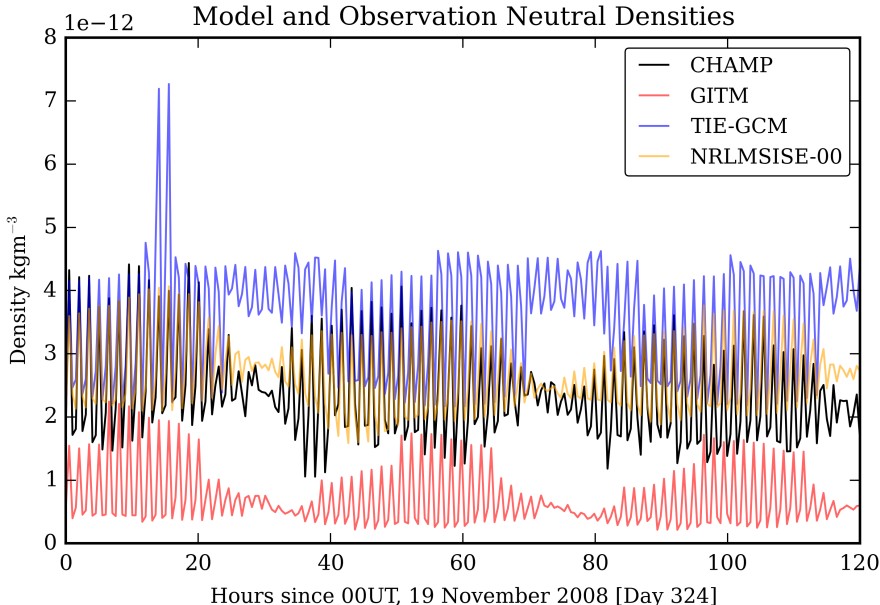

**Figure 4.** CHAMP, GITM, TIE-GCM and NRLMSISE-00 reported neutral densities for the second test scenario (2008; solar minimum).

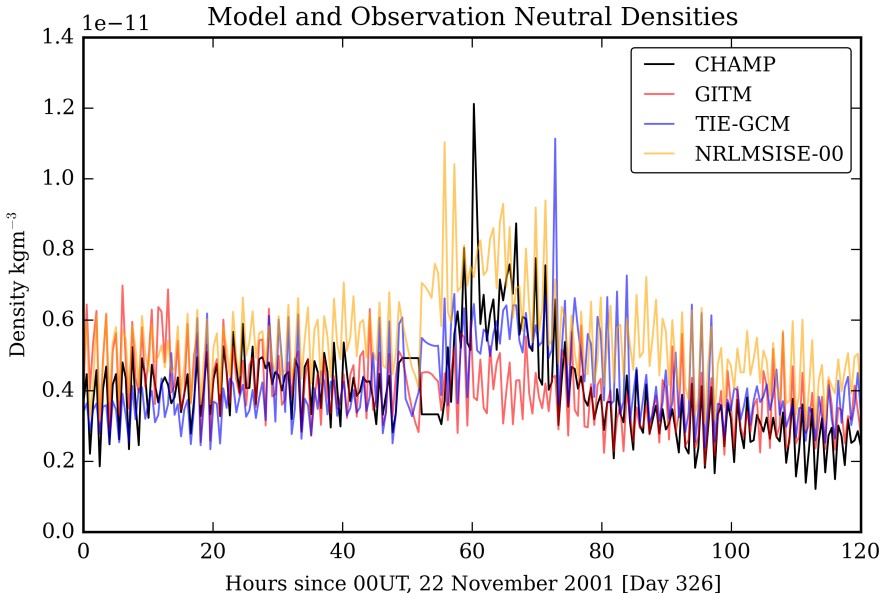

**Figure 5.** CHAMP, GITM, TIE-GCM and NRLMSISE-00 reported neutral densities for the third test scenario (2001; solar maximum).

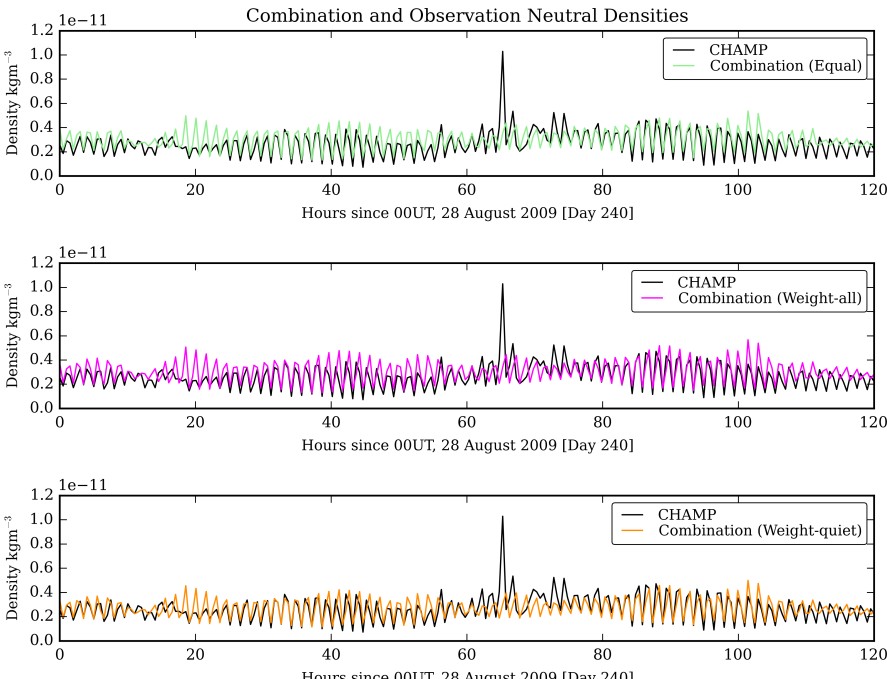

**Figure 6.** Neutral density values of the three MMEs for the first test scenario, equally weighted, quiet-time weighted and all-times weighted.

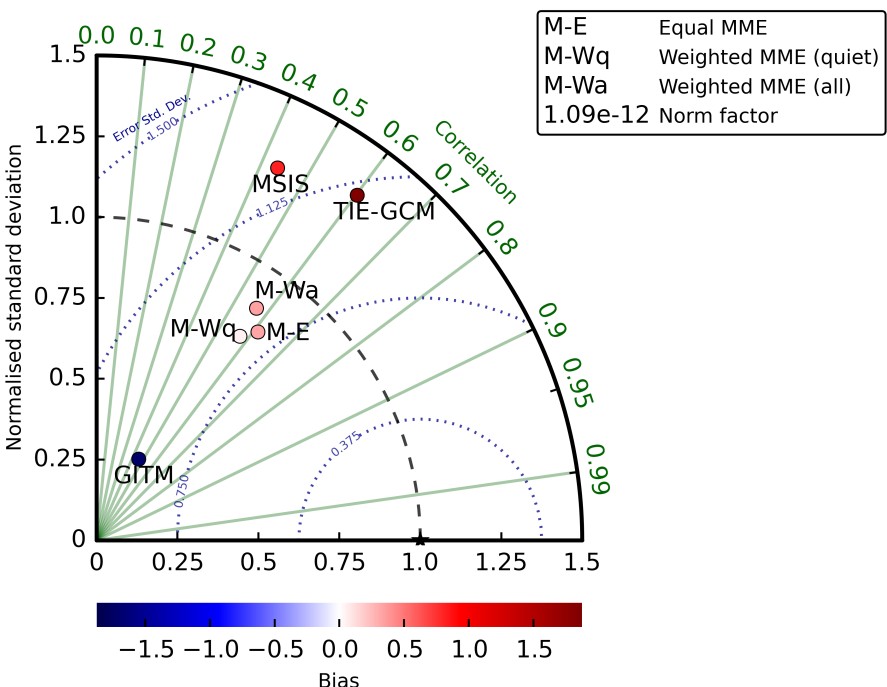

**Figure 7.** Modified Taylor diagram for the three MMEs: equal, quiet-time weighted and all-time weighted as well as GITM, TIE-GCM and NRLMSISE-00 (MSIS) compared to the CHAMP observations for the first test scenario. Details of how to read the diagram are described in Figure 2.

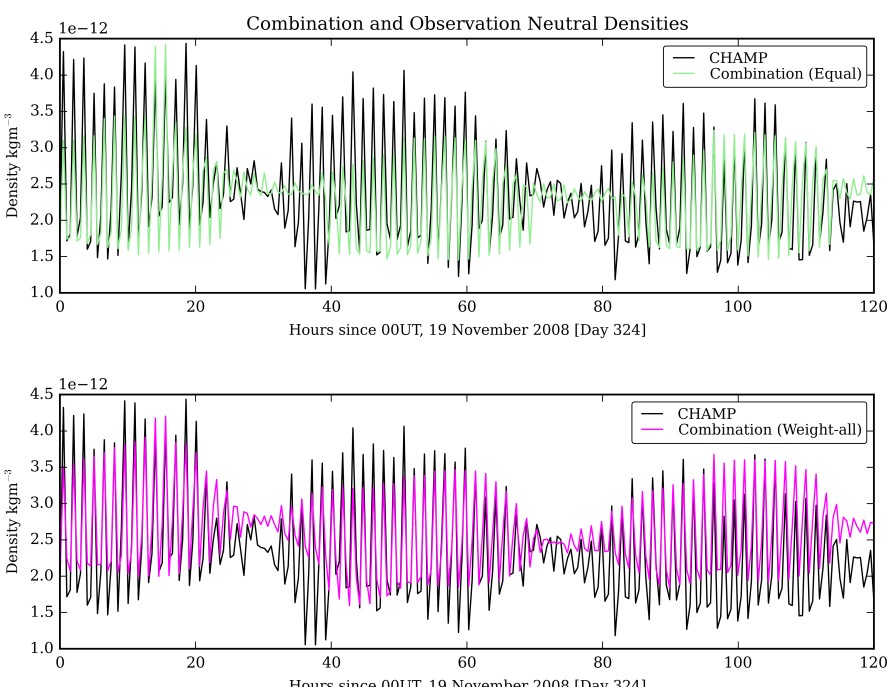

**Figure 8.** Neutral density values of the two MMEs for the second test scenario, equally weighted and all-times weighted.

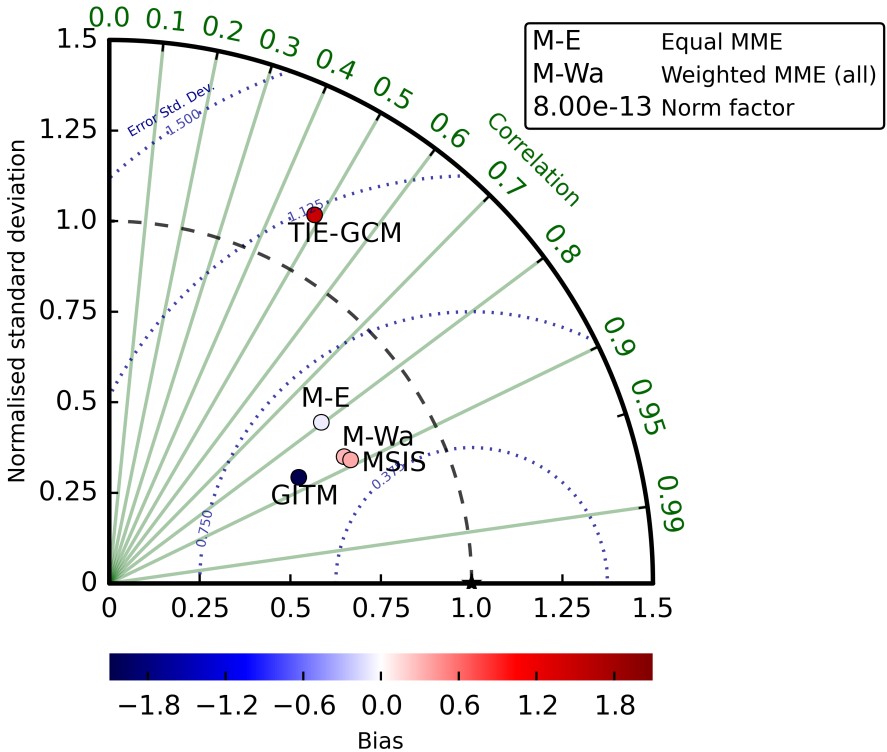

**Figure 9.** Modified Taylor diagram for the two MMEs: equal and all-time weighted as well as GITM, TIE-GCM and NRLMSISE-00 (MSIS) compared to the CHAMP observations for the second test scenario. Details of how to read the diagram are described in Figure 2.

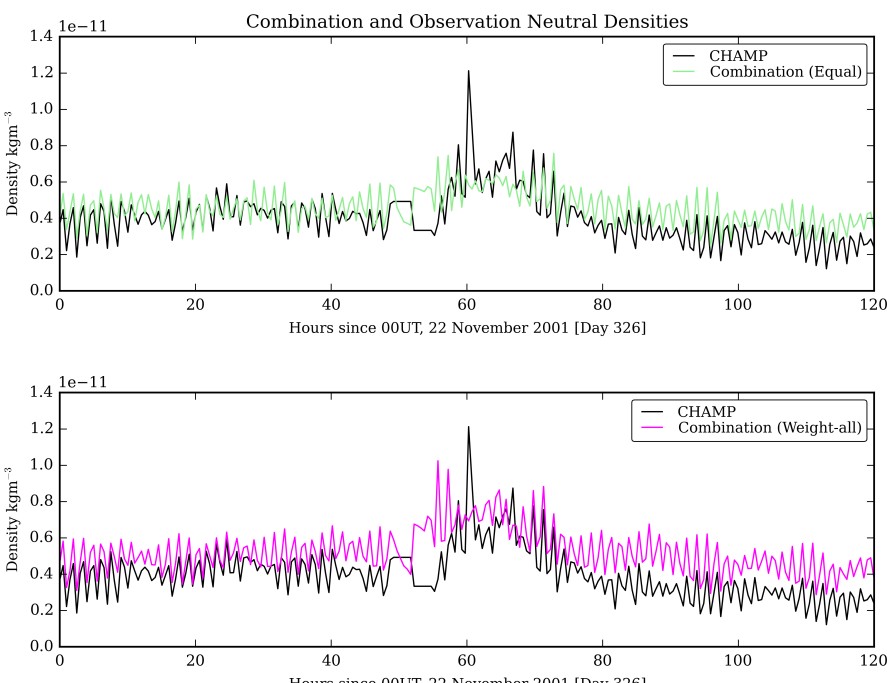

**Figure 10.** Neutral density values of the two MMEs for the third test scenario, equally weighted and all-times weighted.

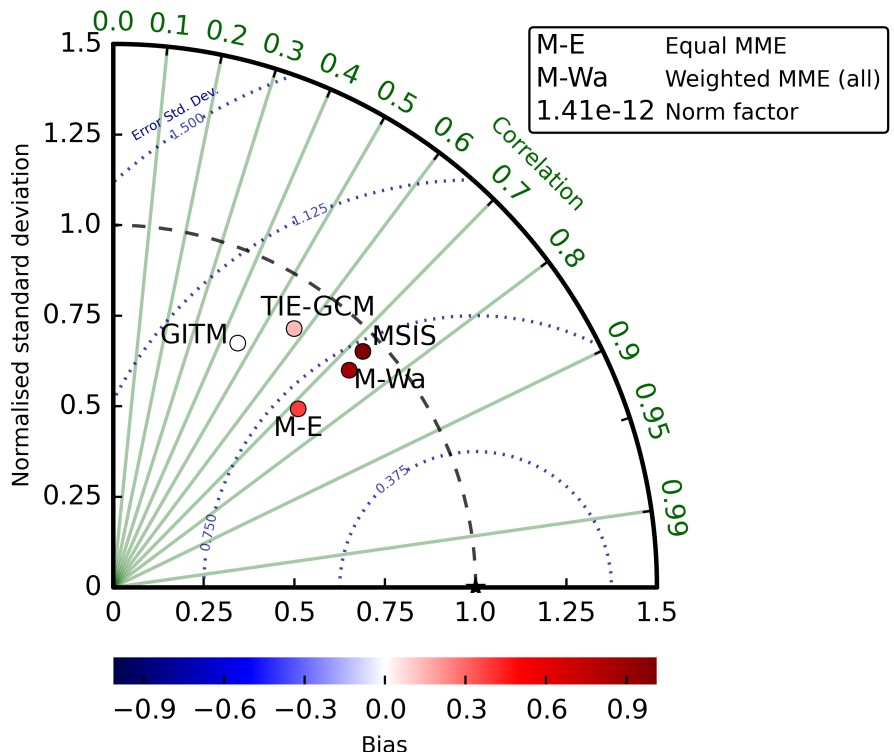

**Figure 11.** Modified Taylor diagram for the two MMEs: equal and all-time weighted as well as GITM, TIE-GCM and NRLMSISE-00 (MSIS) compared to the CHAMP observations for the third test scenario. Details of how to read the diagram are described in Figure 2.

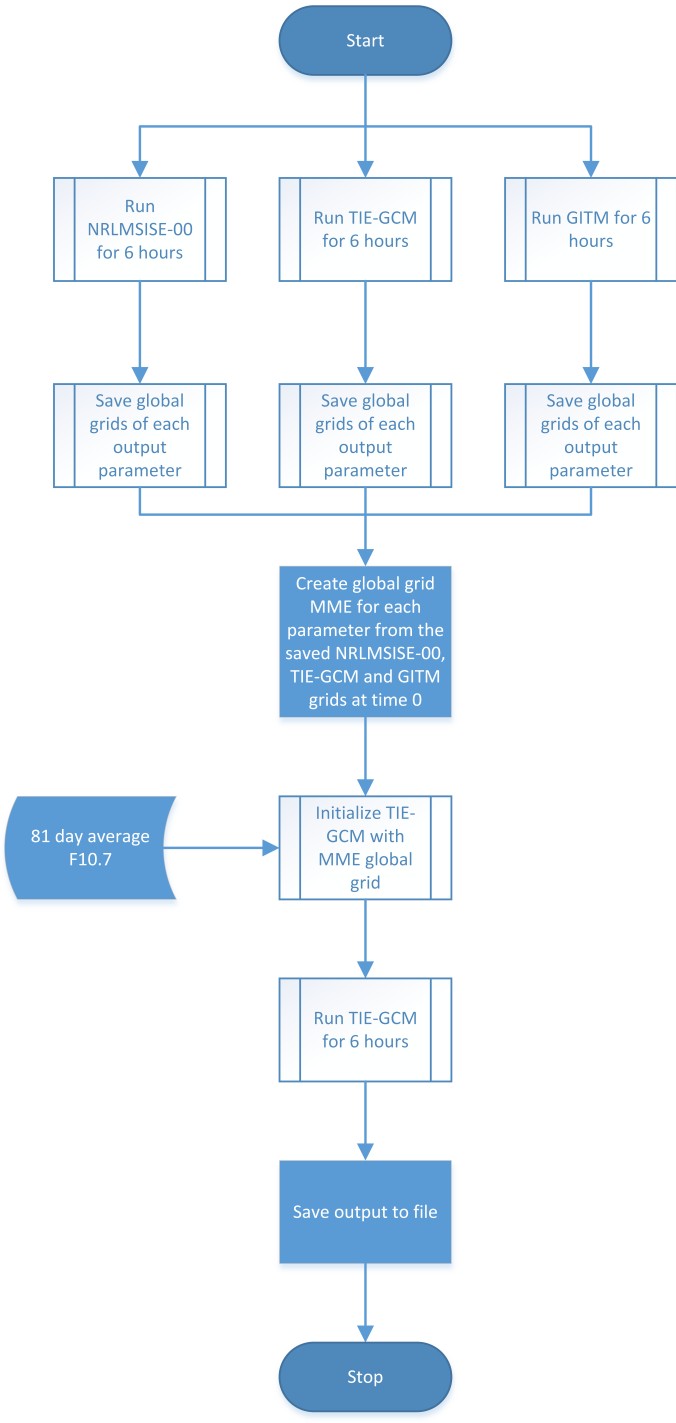

**Figure 12.** Flow chart of the procedure for running TIE-GCM using the MME as its initial conditions for a six hour forecast.

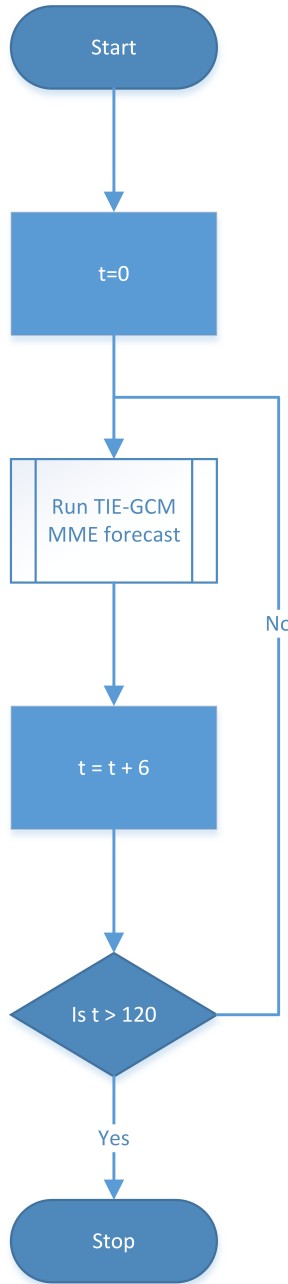

**Figure 13.** Procedure for finding the TIE-GCM forecast using an MME as its initial conditions. The "run TIE-GCM MME forecast" process refers to the procedure described in Figure 12.

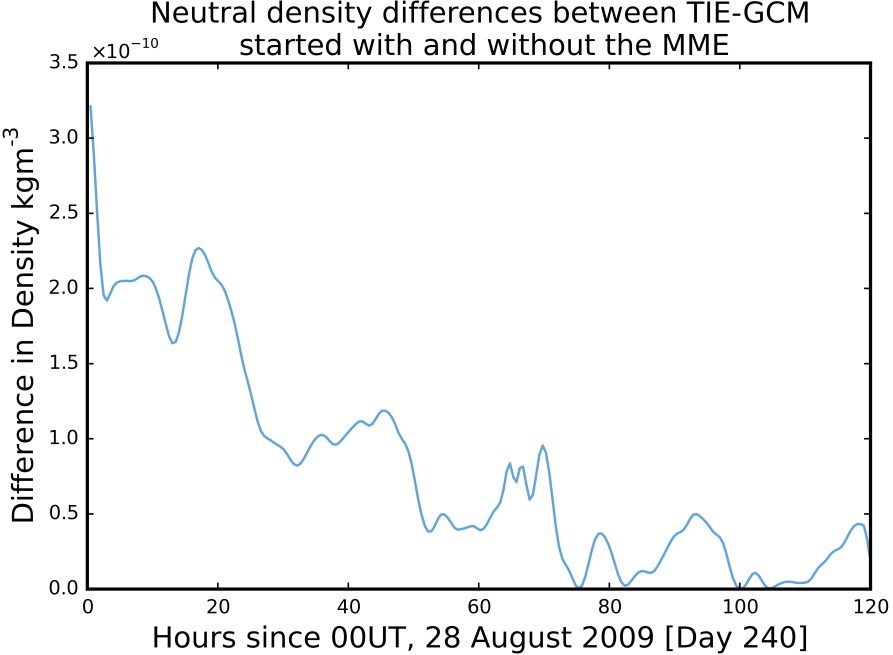

**Figure 14.** Differences between the standard TIE-GCM model run and TIE-GCM ran using the MME as its initial conditions (at time 0). It can be seen that it takes over 70 hours for the models to start to converge again.

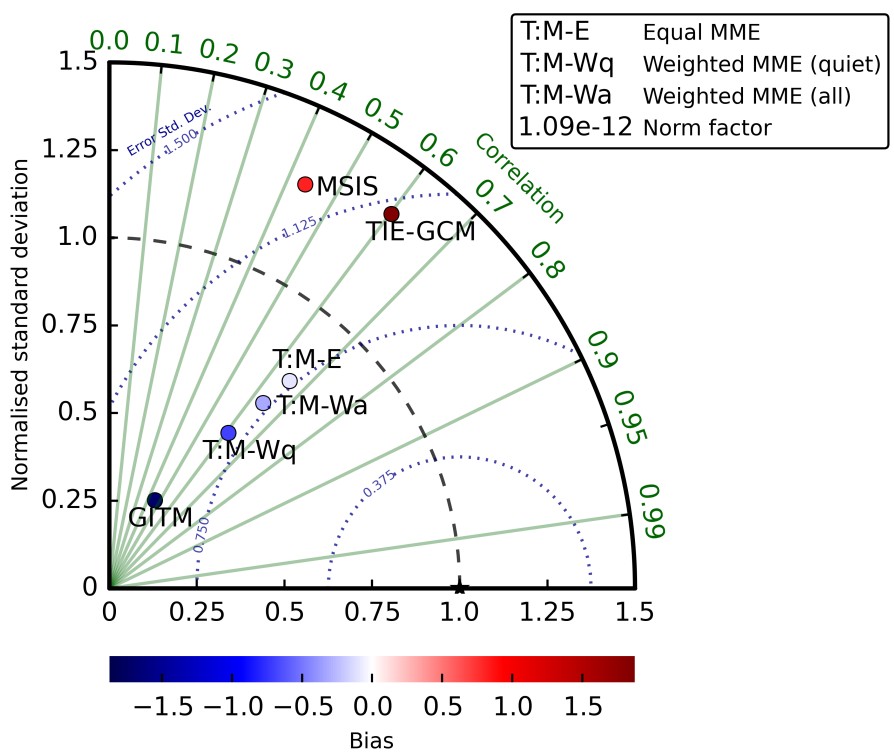

**Figure 15.** Modified Taylor diagram for NRLMSISE-00 (MSIS), TIE-GCM, GITM and for TIE-GCM using the MMEs (equal, quiet-time weighted and all-time weighted) for its initial conditions every six hours, compared with the CHAMP observations. Details of how to read the diagram are described in Figure 2.

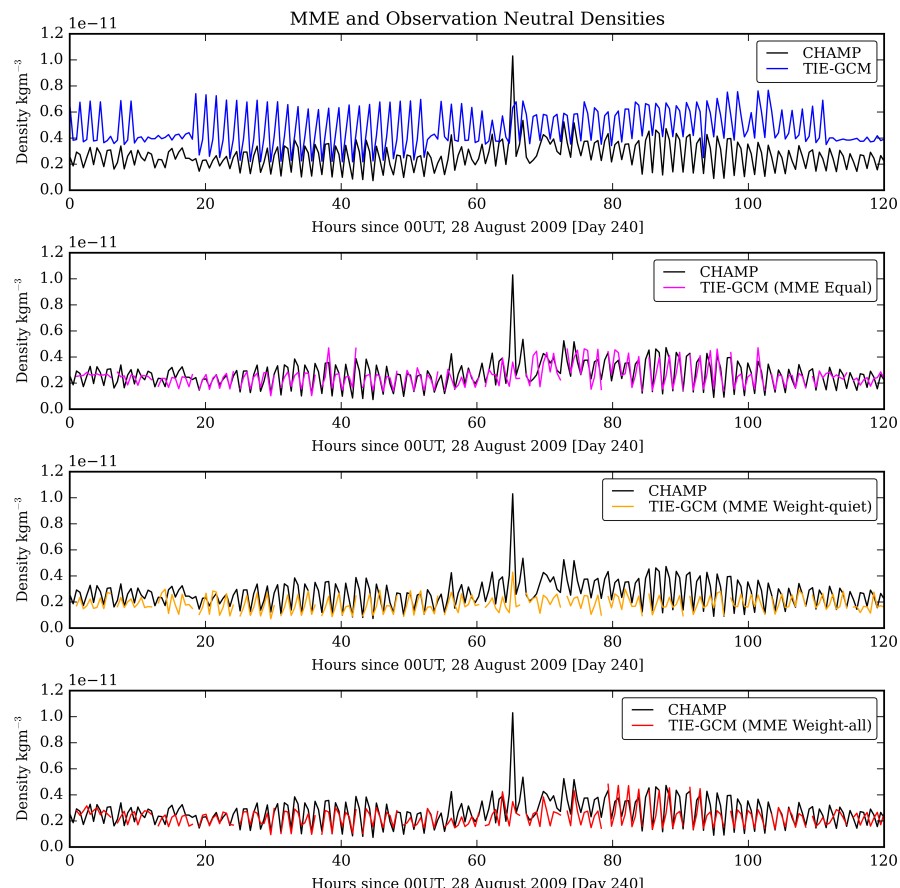

**Figure 16.** Top panel shows the neutral density from the CHAMP observations and the original TIE-GCM run. The subsequent panels then show the CHAMP observations with each of the new TIE-GCM outputs using the MMEs as the initial conditions every six hours.

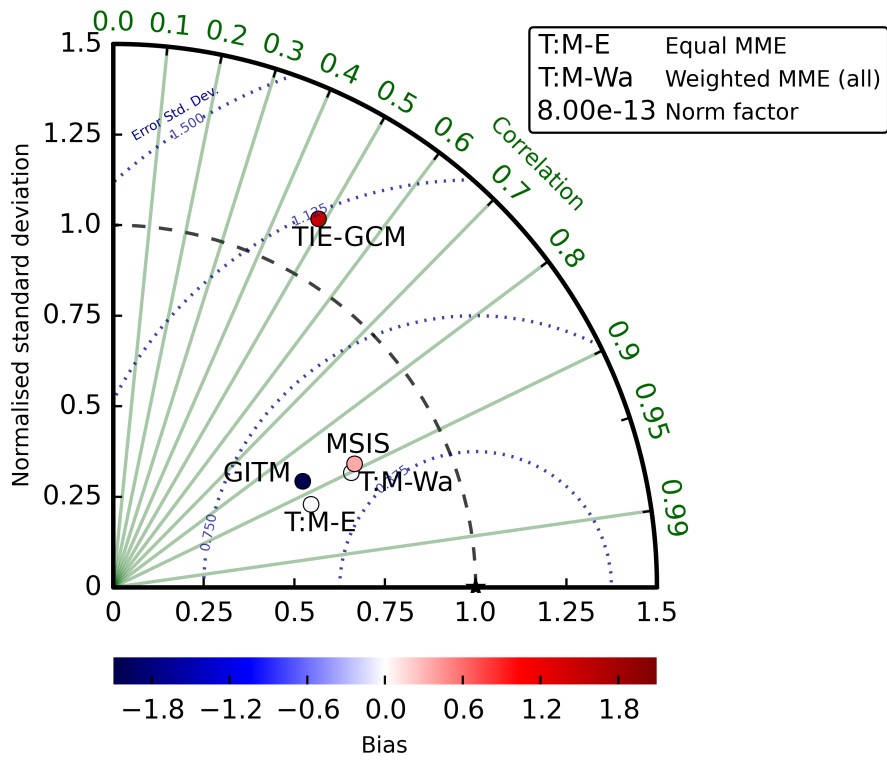

**Figure 17.** Modified Taylor diagram for NRLMSISE-00 (MSIS), TIE-GCM, GITM and for TIE-GCM using the MMEs (equal and all-time weighted) for its initial conditions every six hours, compared with the CHAMP observations. Details of how to read the diagram are described in Figure 2.

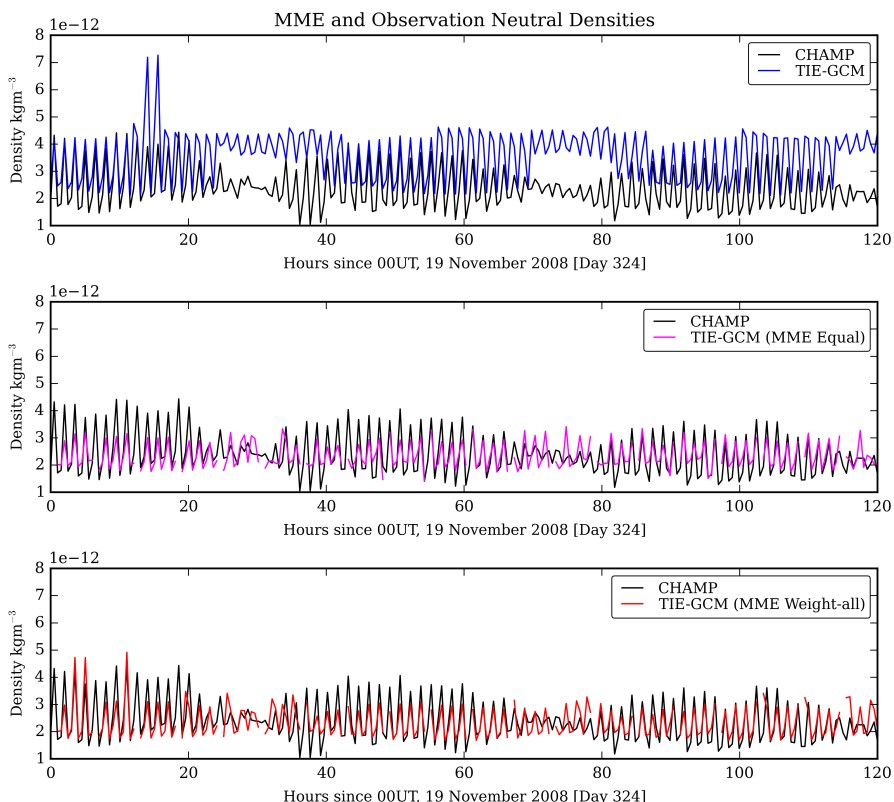

**Figure 18.** Top panel shows the neutral density from the CHAMP observations and the original TIE-GCM run. The subsequent panels then show the CHAMP observations with each of the new TIE-GCM outputs using the MMEs as the initial conditions every six hours.

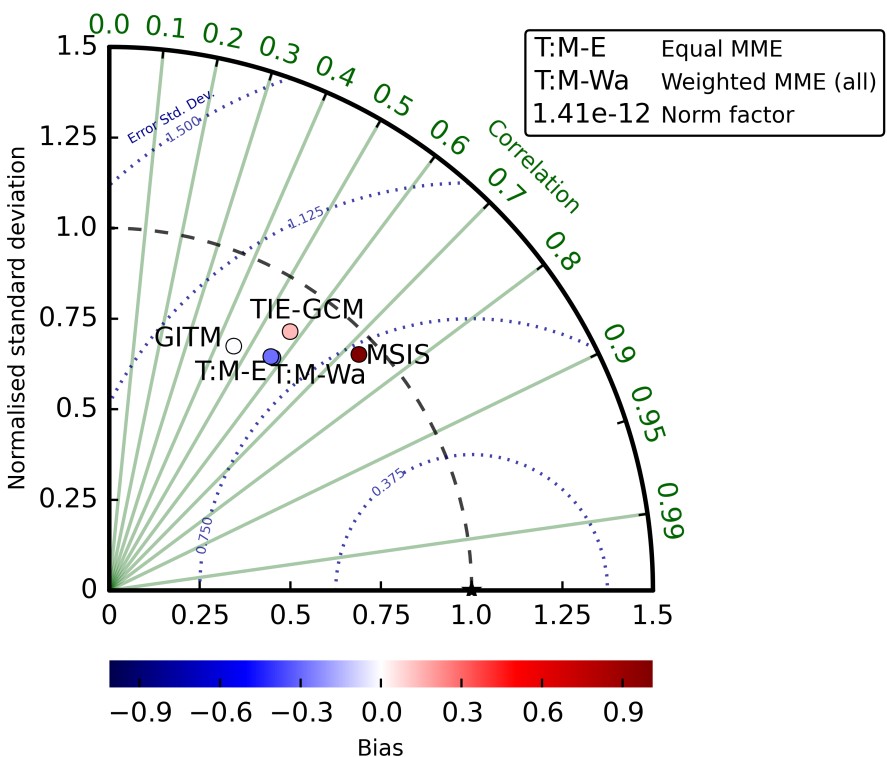

**Figure 19.** Modified Taylor diagram for NRLMSISE-00 (MSIS), TIE-GCM, GITM and for TIE-GCM using the MMEs (equal and all-time weighted) for its initial conditions every six hours, compared with the CHAMP observations. N.B. The two markers for the MMEs overlap each other. Details of how to read the diagram are described in Figure 2.

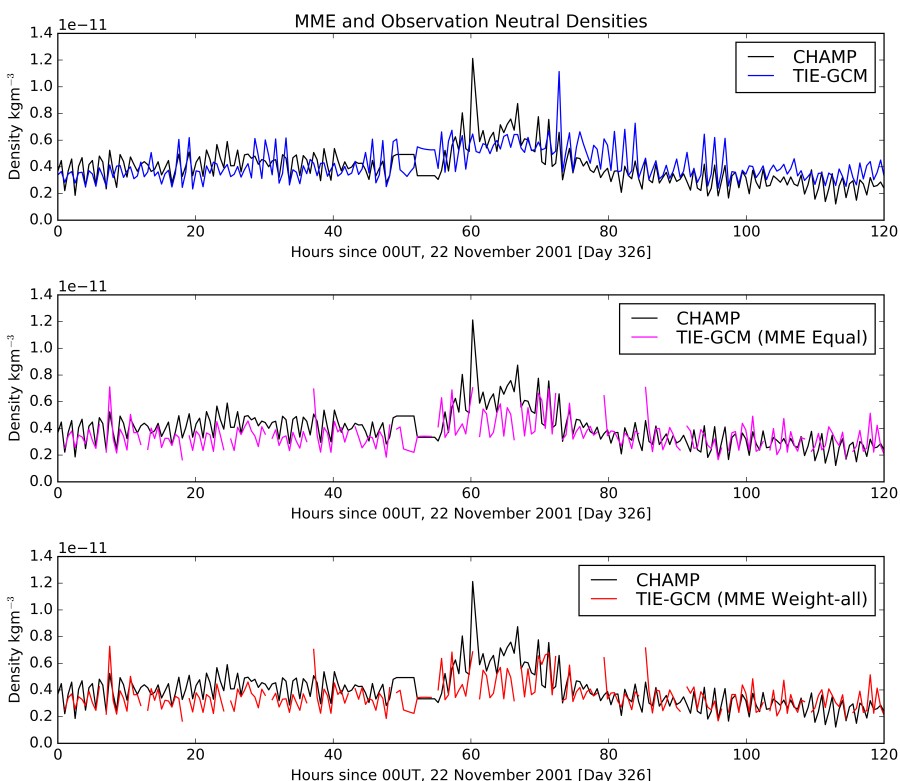

**Figure 20.** Top panel shows the neutral density from the CHAMP observations and the original TIE-GCM run. The subsequent panels then show the CHAMP observations with each of the new TIE-GCM outputs using the MMEs as the initial conditions every six hours.