# Peer review of "Improved Forecasting of Thermospheric Densities using Multi-Model Ensembles"

_Geoscientific Model Development, 2015_

## Referee Comment (RC1) · Anonymous Referee #1 · 9 Mar 2016

Review comments to gmd-2015-203:

This paper presented the application of Multi-Model Ensembles (MME) for the thermospheric density forcasting. First, the neutral density from three individual models has been compared with CHAMP satellite observations. The MME has been constructed from three models, which shows a significant improvement compared with individual models. TIE-GCM has also been run for forecasting and initialized with MME every 6 hours The results are very interesting and relevant to the current topics. The paper is suitable for the publication of GMD. However, some methodology is not very clear and the clarification is needed.

(1) Line 202: "The model all perform very similarly": To my eyes, the model results in Figure 5 are quite different, especially the variation of neutral density during the

storm period, even the standard deviation and correlation are similar. Does it indicate the limitation of using standard deviation and correlation to judge the performance of simulation?

(2) Line 205-209 and Line 251: the performance difference between the third scenario and the first two is mainly explained as the consequence of the solar activity variation. However, F10.7 actually represents the solar irradiation worse during last extremely quiet solar minimum in 2008 and 2009 than the solar maximum in 2001. It is not clear that the comparison is necessarily worse in the solar maximum when the models are driven by F10.7. Meanwhile, the third scenario includes a much larger geomagnetic storm than the other two, which may also contribute the performance difference in addition to the solar activity change.

(3) Line 296: "Using the MME as the initial condition in TIE-GCM . . .": The terminology of "initial condition" is confusing. Typically, the initial condition is a one-time thing for the simulation, which is used at the beginning of the simulation period. What has actually been done in this study is to retune TIE-GCM to MME every 6 hours, which is probably different from the initial conditions people usually talk about.

(4) Table 2: The weights are quite different from one scenario to another. It may indicate that the weight MME may not be applicable for forecasting using the weight calculated from historic events.

(5) The abstract needs to represent the content of the paper better by including more information about the approach and main conclusion.

(6) Figure 20: the label shows the period of Nov. 2008, which is the second scenario.

(7) Line 252: t → It

Please also note the supplement to this comment:
http://www.geosci-model-dev-discuss.net/gmd-2015-203/gmd-2015-203-RC1-

supplement.pdf

---

## Referee Comment (RC2) · Anonymous Referee #2 · 9 May 2016

This paper describes a multi-model ensemble (MME) of thermosphere models. The MME was used to simulate three events. Three ways of combining the three models were used: (1) no weighting, (2) weighting by the error between the model and the data, and (3) weighting by the error, but only counting quiet times. The easiest method, no weighting, seems like it worked the most reliably. Then, a single model was initialized with the MME and used to forecast with good success.

Comments:

Line 164: The CHAMP data was probably taken more often that 45 seconds, but then was averaged to 3 degrees in latitude, which is about 45 seconds.

Lines 165-175: There is a change in tense in this paragraph compared to the rest of the paper that should be corrected.

[Figure]
Line 178: Trilinear interpolation was mentioned. Does this mean the model files were output at the exact correct time? How often were the model output files written?

Line 252: The "I" is missing in "It"

Line 258: Comma after "conditions"

Line 261: Comma after "possible"

Line 262-263: Should use "e.g.," instead of "i.e."

Line 267: Comma after "models"

Line 270: Need should be "needed". Comma after "grids"

Line 274: Comma after "run", "grid" should be "state-vector", and then "conditions" should be "condition"

Line 275: "this" should be "the six-hour period"

Line 276: Comma after "period"

Line 278: The argument here that this is a "true forecast" is not really true at all, since it would imply that you are running all three models with real inputs/drivers. What really should be done is that all three models should be run in a "predictive" mode for 6 hours, then the MME should occur and all three should be re-initialized. Then the sequence should start again. But, as it is now, the MME is like a "truth" simulation and you are always bringing the TIEGCM back up to the truth. This is not like how it would be done in real predictive mode.

Line 284: "conditions" to "condition"

Line 285: "combine" to "converge"

Line 288: Change the rest of the sentence starting with "the two models..." to "the two models, started with different initial conditions, are decreasing towards zero, as expected, but it takes approximately 70 hours to read these levels." Really, a much

better way to do this would be to fit an exponential decay to this curve and report the e-folding time. It should never really reach zero.

Line 305 and around there: It seems like the original MME - combining the 3 models - does better than the TIEGCM initialized with the MME. Why not run all 3 models and use an MME for the forecast also?

---

## Author Comment (AC1) · 17 May 2016

Many thanks for your review of our paper. We have responded to your comments in order below:

"(1) Line 202: "The model all perform very similarly": To my eyes, the model results in Figure 5 are quite different, especially the variation of neutral density during the storm period, even the standard deviation and correlation are similar. Does it indicate the limitation of using standard deviation and correlation to judge the performance of simulation?"

We agree that there are greater differences in the model performance than was specifed here. In particular the differences during the storm period. We think that the models perform 'more' similarly in this scenario than the others, but perhaps not to

the extent that we have made out. The text has been updated to reflect this fact.

"(2) Line 205-209 and Line 251: the performance difference between the third scenario and the first two is mainly explained as the consequence of the solar activity variation. However, F10.7 actually represents the solar irradiation worse during last extremely quiet solar minimum in 2008 and 2009 than the solar maximum in 2001. It is not clear that the comparison is necessarily worse in the solar maximum when the models are driven by F10.7. Meanwhile, the third scenario includes a much larger geomagnetic storm than the other two, which may also contribute the performance difference in addition to the solar activity change."

Thank you for highlighting this point. Although it is known that F10.7 did a worse job at representing solar irradiance between 2008 and 2009 I do still think that this is the cause for the main differences in model variation between the test scenarios. The poor job that F10.7 did could help explain something about the errors between the models and observations. However since all the models used the same F10.7 value as a driver this "lack of performance" shouldn't have any impact on the model spread. Similarly the larger geomagnetic storm in the third test scenario could have an impact, but probably only during the storm itself. Before the storm hits is when the the model spread is at its least and is only being driven by the F10.7 value. I do think though that it is important to highlight the F10.7 issues during this time, as such we have commented on them in the paper and included a further reference.

"(3) Line 296: "Using the MME as the initial condition in TIE-GCM . . .": The terminology of "initial condition" is confusing. Typically, the initial condition is a one-time thing for the simulation, which is used at the beginning of the simulation period. What has actually been done in this study is to retune TIE-GCM to MME every 6 hours, which is probably different from the initial conditions people usually talk about."

We appreciate that the terminology can be confusing here. To try and address any confusion we have replaced this 'initial condition' discussion (and in other places in

the paper) with the idea of using the MME densities as the starting densities for TIE-GCM..."

"(4) Table 2: The weights are quite different from one scenario to another. It may indicate that the weight MME may not be applicable for forecasting using the weight calculated from historic events."

This is a very useful comment and we have now commented in the paper about the large differences in weightings from one scenario to another.

"(5) The abstract needs to represent the content of the paper better by including more information about the approach and main conclusion."

We have updated the abstract to, without a huge increase in the number of words, better explain the content of the paper.

"(6) Figure 20: the label shows the period of Nov. 2008, which is the second scenario."

This has now been updated.

"(7) Line 252: t → It"

Corrected.

---

## Author Comment (AC2) · 17 May 2016

Many thanks for the review of our paper. We have replied to your specific comments below:

"Line 164: The CHAMP data was probably taken more often that 45 seconds, but then was averaged to 3 degrees in latitude, which is about 45 seconds."

Thank you for this. It would seem the accelerometer data was collected at rate of 1 Hz, and then averaged as you suggested. I have updated the text to reflect this.

"Lines 165-175: There is a change in tense in this paragraph compared to the rest of the paper that should be corrected. "

Corrected.

[Figure]

"Line 178: Trilinear interpolation was mentioned. Does this mean the model files were output at the exact correct time? How often were the model output files written?"

It was an oversight to not include this information. The model output files were written every 30 minutes and the closest CHAMP time was taken. Since there is CHAMP data every 45 seconds, the closest matching time is always within one internal model timestep. As such it was decided that this would have little impact on the results. However this information should be included in the text which I have now updated.

"Line 252: The "I" is missing in "It" "

Corrected.

"Line 258: Comma after "conditions" "

Added.

"Line 261: Comma after "possible" "

Added.

"Line 262-263: Should use "e.g.," instead of "i.e." "

Corrected.

"Line 267: Comma after "models" "

Added.

"Line 270: Need should be "needed". Comma after "grids" "

Corrected.

"Line 274: Comma after "run", "grid" should be "state-vector", and then "conditions" should be "condition" "

Corrected

"Line 275: "this" should be "the six-hour period" "

Corrected.

"Line 276: Comma after "period" "

Added

"Line 278: The argument here that this is a "true forecast" is not really true at all, since it would imply that you are running all three models with real inputs/drivers. What really should be done is that all three models should be run in a "predictive" mode for 6 hours, then the MME should occur and all three should be re-initialized. Then the sequence should start again. But, as it is now, the MME is like a "truth" simulation and you are always bringing the TIEGCM back up to the truth. This is not like how it would be done in real predictive mode. "

I believe in this case that what we are describing is a true forecast. TIE-GCM, initialised with the equally weighted MME, is run forward for 6 hours with no external driver information (since this wouldn't be available in reality). This is how the system could be run operationally. The MME is created using historic model information (up to 'now' in a real system) and then propagated forward. The MME forecast is then compared to the observations over that time period for us to judge how successful the approach was. The mean square error weighted MME however isn't entirely a true forecast, since the weighting is created using data from the entire time period. This information wouldn't be available in a real predictive mode. We have noted this in the text.

"Line 284: "conditions" to "condition" "

Corrected.

"Line 285: "combine" to "converge" "

Corrected.

"Line 288: Change the rest of the sentence starting with "the two models..." to "the

two models, started with different initial conditions, are decreasing towards zero, as expected, but it takes approximately 70 hours to read these levels." Really, a much better way to do this would be to fit an exponential decay to this curve and report the e-folding time. It should never really reach zero. "

I've updated the text as you suggested and also included the e-folding time.

"Line 305 and around there: It seems like the original MME - combining the 3 models - does better than the TIEGCM initialized with the MME. Why not run all 3 models and use an MME for the forecast also?"

An MME could be used for the forecast by running each of the models forward and then combining the outputs (the models would have to use estimated driving parameters). However such an approach is expected to have worse performance long-term since if you have a poor specification of the densities in the first place propagating them forwards with the models would result in poor forecasts. The point of the MME should be to reduce the uncertainties in the initial conditions for better forecasts. An approach which could yield an improvement would be to use the MME as the initial conditions to multiple models, run each forward for the forecast window, and then create a further MME by combining the outputs. The final MME would then be the best estimate for the forecast. However this can not yet be achieved since we have not got GITM running using the MME as the initial conditions. This is something we are working on.